# Establishing SW1353 Chondrocytes as a Cellular Model of Chondrolysis

**DOI:** 10.3390/life11040272

**Published:** 2021-03-25

**Authors:** Kok-Lun Pang, Yoke Yue Chow, Lek Mun Leong, Jia Xian Law, Norzana Abd Ghafar, Ima Nirwana Soelaiman, Kok-Yong Chin

**Affiliations:** 1Department of Pharmacology, Faculty of Medicine, Universiti Kebangsaan Malaysia, Kuala Lumpur 56000, Malaysia; pangkoklun@ukm.edu.my (K.-L.P.); imasoel@ppukm.ukm.edu.my (I.N.S.); 2Department of Orthopaedic and Trauma Medicine, Deanery of Clinical Sciences, The University of Edinburgh, Edinburgh EH16 4SB, UK; s2004323@ed.ac.uk; 3Prima Nexus Sdn. Bhd., Kuala Lumpur 50470, Malaysia; raymond@primanexus.com.my; 4Department of Biomedical Science, Faculty of Science, Lincoln University College, Petaling Jaya 47301, Malaysia; 5Centre for Tissue Engineering and Regenerative Medicine, Universiti Kebangsaan Malaysia Medical Centre, Kuala Lumpur 56000, Malaysia; lawjx@ppukm.ukm.edu.my; 6Department of Anatomy, Faculty of Medicine, Universiti Kebangsaan Malaysia, Kuala Lumpur 56000, Malaysia; norzana@ukm.edu.my

**Keywords:** interleukin-1 beta, monosodium iodoacetate, hydrogen peroxide, oxidative stress, inflammation

## Abstract

Osteoarthritis (OA) is the most common degenerative joint disease characterised by chondrocyte cell death. An in vitro model of chondrocyte cell death may facilitate drug discovery in OA management. In this study, the cytotoxicity and mode of cell death of SW1353 chondrocytes treated with 24 h of OA inducers, including interleukin-1β (IL-1β), hydrogen peroxide (H_2_O_2_) and monosodium iodoacetate (MIA), were investigated. The microscopic features, oxidative (isoprostane) and inflammatory markers (tumour necrosis factor-alpha; TNF-α) for control and treated cells were compared. Our results showed that 24 h of H_2_O_2_ and MIA caused oxidative stress and a concentration-dependent reduction of SW1353 cell viability without TNF-α level upregulation. H_2_O_2_ primarily induced chondrocyte apoptosis with the detection of blebbing formation, cell shrinkage and cellular debris. MIA induced S-phase arrest on chondrocytes with a reduced number of attached cells but without significant cell death. On the other hand, 24 h of IL-1β did not affect the cell morphology and viability of SW1353 cells, with a significant increase in intracellular TNF-α levels without inducing oxidative stress. In conclusion, each OA inducer exerts differential effects on SW1353 chondrocyte cell fate. IL-1β is suitable in the inflammatory study but not for chondrocyte cell death. H_2_O_2_ and MIA are suitable for inducing chondrocyte cell death and growth arrest, respectively.

## 1. Introduction

Osteoarthritis (OA) is a degenerative joint disease commonly found among the elderly [1]. The exact aetiology of OA has been unclear until now [2]. Several risk factors like age (>50 years), gender (female), obesity, diabetes, past joint trauma, strenuous exercise, and genetic susceptibility could increase the risk of OA [2,3]. Currently, there is no effective treatment that could delay, stop or reverse the cartilage destruction due to an incomplete understanding of the molecular mechanisms of OA initiation and progression [2,4]. The hypocellularity of articular cartilage due to chondrocyte cell death is one of the hallmarks of OA initiation [5]. In the early stage of OA, chondrocytes are damaged by mechanical stress, oxidative stress or inflammation [6]. They have limited capacity to recover from these assaults due to the avascular and alymphatic nature of articular cartilage [7,8]. The damaged chondrocytes may experience hypertrophy-like changes and lose the ability to synthesise proteoglycan [9]. Eventually, the cells will die due to an impaired reparative mechanism, leading to hypocellularity, matrix depletion and degradation of cartilage [5]. Additionally, the leaked cellular content and degradation products from dead cells will further exacerbate the inflammatory process by enhancing the release of proinflammatory cytokines like prostaglandins, interleukin-1β (IL-1β) and tumour necrosis factor-α (TNF-α) as well as proteolytic enzymes like aggrecanases, collagenases and matrix metalloproteinases (MMPs) from cartilage cells and immune cells [7]. Ultimately, this vicious cycle causes the progressive loss of cartilage, joint space narrowing, subchondral bone remodelling, synovium inflammation and fibrosis, which will be manifested along with the progression of OA [10].

Several OA models are currently used in OA research, including chondrocyte cell lines, primary articular chondrocytes and OA animals [10,11,12]. Primary articular chondrocytes from OA patients are undeniably the most representative cell culture model of OA. However, the use of primary chondrocytes is limited by ethical issues, the availability of cartilages and low yield from the isolation procedures [13]. Besides, primary chondrocytes have the disadvantages of heterogeneity in cell types, loss of phenotype and limited proliferative capacity during ex vivo culturing [14]. The use of commercially available chondrocyte cell lines could overcome these limitations because they are readily available, cost-effective, technically less challenging to culture and without significant ethical issues compared with primary chondrocytes. Several human chondrocyte cell lines have been used as in vitro OA models, including SW1353 chondrocytes [15,16,17], CHON-001 cells [18,19], CHON-002 cells [20], and C-28/I2, T/C-28a2 and T/C-28a4 chondrocytes [21,22].

Chondrosarcoma cell line SW1353 is one of the most common and established cell lines used as in vitro OA model [15,16,17,23]; its first use in an interleukin (IL)-1-induced inflammation model was recorded in 1987 [24]. To mimic the initiation and pathogenesis of OA, SW1353 chondrocytes are generally stimulated with proinflammatory cytokines like IL-1β or TNF-α to trigger the inflammatory response and catabolic profile, marked by upregulation of MMPs and cytokine production [25,26]. However, IL-1β-stimulated SW1353 cells were reported to express similar but weaker chondrogenic and catabolic profiles compared with primary human chondrocytes [25]. The cytotoxicity of IL-1β on primary human articular chondrocytes [27] and chondrocyte cell lines upon 24 to 48 h treatment has been well reported [28,29]. IL-1β stimulation does not directly damage chondrocytes but acts via the mitochondrial dysfunction and upregulation of reactive oxygen species (ROS) to execute chondrocyte cell death [30]. Additionally, IL-1β has also been used in in vivo OA models via intra-articular injection [12].

Inflammation- or age-related oxidative stress is crucial in OA progression, where chondrocytes can express the catabolic profile upon oxidative damage [31]. The oxidant properties of hydrogen peroxide (H_2_O_2_), a non-radical ROS, has been widely reported [32]. H_2_O_2_ is membrane-permeable [33], whereby it can cross the plasma membrane and cause oxidative damage and cell death [34]. The cytotoxicity of H_2_O_2_ has been reported in primary articular chondrocytes from animals and humans [34,35], human C28/12 chondrocytes [36] and SW1353 chondrocytes [15,37,38,39]. The functionality and telomere stability of surviving chondrocytes were also affected [40]. Exogenous H_2_O_2_ and ROS inducers like menadione are commonly used in the in vitro approach to induce oxidative stress in chondrocytes [41,42]. Additionally, intra-articular injection of H_2_O_2_ is rarely used but can possibly induce OA in animals [12]. Comparatively, chemicals like monosodium iodoacetate (MIA) are more frequently used to mimic OA in animal models [16,43].

Articular chondrocytes acquire oxygen and nutrients from synovial fluid via diffusion due to the anatomically avascular and alymphatic structures of the cartilage tissues [7,8,31]. The oxygen tension in cartilage tissues is relatively lower (<10%), and the energy production in chondrocyte is mainly via glycolysis [44]. Due to this reason, chondrocytes are relatively sensitive to glycolysis inhibition compared with other cell types. MIA is a known glycolysis inhibitor, which binds irreversibly with the sulfhydryl group of glyceraldehyde-3-phosphate dehydrogenase and then inhibits its activity [45]. Additionally, MIA also upregulates ROS production, probably via the glutathione (GSH) depletion and/or mitochondrial dysfunction [16,43,46]. MIA-mediated oxidative stress and energy depletion ultimately cause the death of primary articular chondrocytes [46] and chondrocyte cell lines [16,17,43]. Intra-articular injection of MIA causes the cytotoxicity in chondrocytes, leading to cartilage degradation and subchondral bone alterations, which mimics the pathogenesis of OA [12,43].

SW1353 chondrocytes are among the most established cell lines in in vitro OA studies. Despite the prevalent use of IL-1β, H_2_O_2_ and MIA in SW1353 chondrocyte models, no direct comparison among these inducers of SW1353 cell death has been made. This research gap will impede the selection of an appropriate inducer in in vitro OA studies. So far, only a study by Martin et al. comparing the differential catabolic effects of IL-1β and H_2_O_2_ on juvenile bovine chondrocytes is available [47]. The differential effects of these common OA inducers on chondrocyte cell death are largely unknown. Therefore, in this study, we investigated and compared the IL-1β-, H_2_O_2_- and MIA-induced cytotoxicity and mode of cell death in SW1353 chondrocytes. The involvement of oxidative stress and inflammation were also investigated. This is the first study to investigate the differential effects of these OA inducers. Such information would benefit researchers in selecting an appropriate agent for their in vitro OA models.

## 2. Materials and Methods

### 2.1. Materials

MIA was purchased from Sigma-Aldrich (St. Louis, MO, USA), dissolved in dimethyl sulfoxide (DMSO) as a 100 mM stock solution, aliquoted and kept at −80 °C until treatment. Concentrated 30% H_2_O_2_ solution was purchased from Merck Millipore (Merck KGaA, Gernsheim, Germany) and kept at 4 °C until treatment. During treatment, the 30% H_2_O_2_ solution was pre-diluted with sterile phosphate-buffered saline (PBS) in dark conditions to prepare a 20 mM H_2_O_2_ solution, then continued to the second dilution step in culture media to prepare the treatment solution accordingly. Lyophilised recombinant human IL-1β with albumin (catalogue No. 8900SC) was purchased from Cell Signaling Technology, Inc. (Danvers, MA, USA) and reconstituted according to the manufacturer’s instructions. Briefly, 10 μg of IL-1β powder was dissolved with sterile PBS as a 50 μg/mL stock solution at room temperature with 30 min of solubilization and occasional gentle vortexing. The dissolved IL-1β was aliquoted and kept at −80 °C until treatment. The stock solutions of IL-1β, H_2_O_2_ and MIA were always protected from light during preparation, storage and treatment. Other chemicals were purchased from Sigma-Aldrich unless stated otherwise.

### 2.2. Cell Lines

SW1353 chondrocytes were purchased from ATCC (catalogue o. HTB 94; lot No. 63591988; passage number 22) and cultured with high glucose Dulbecco’s modified Eagle’s medium (DMEM)(ThermoFisher Scientific, Waltham, MA, USA; catalogue No. 11965-092) with 4 mM l-glutamine, 10% fetal bovine serum (Thermo Fisher Scientific, Waltham, MA, USA; catalogue No. 10270-106) and 1% penicillin–streptomycin antibiotic solution (Thermo Fisher Scientific, Waltham, MA, USA; catalogue No. 15140-122) at 37 °C and 5% carbon dioxide conditions. These cells were sub-cultured every 2–3 days to maintain optimal cell growth. The cells within the first 3 to 10 passages after being revived from cryopreservation were used in the experiment.

### 2.3. Cytotoxicity Assay

CellTiter 96 AQueous One solution (Promega, Fitchburg, WI, USA) containing 3-(4,5-dimethylthiazol-2-yl)-5-(3-carboxymethoxyphenyl)-2-(4-sulfophenyl)-2H-tetrazolium salt (MTS) was used to determine the cytotoxicity of IL-1β, H_2_O_2_ and MIA on SW1353 chondrocytes as previously described, with modifications [48,49]. Briefly, 100 µL of 5 × 10^4^ cells/mL of SW1353 chondrocytes was seeded in a 96-well plate with complete media for 24 h. On the next day, all the media were then replaced with 100 µL of complete media with a series of concentration of either IL-1β, H_2_O_2_ and MIA for another 24 h. DMSO (0.05%) was used as vehicle control (VC). At the end of the treatment, 20 µL of MTS reagent was added to each well and the plate was further incubated for another 1 h at 37 °C. The absorbance of each well was then measured at 490 nm using a Multiskan GO microplate reader (Thermo Fisher Scientific, Vantaa, Finland). The viability of cells was calculated as a percentage with the optical density (OD) value of the VC as the denominator. The values of the 25% and 50% maximal inhibitory concentration (IC_25_ and IC_50_) were determined for subsequent experiments.

### 2.4. Annexin V–Fluorescein Isothiocyanate/Propidium Iodide Dual-Labelling Assay

The mode of cell death, either via apoptosis or necrosis, was determined according to the plasma membrane externalization of phosphatidylserine and its integrity [50]. This assay was conducted in accord with a previous study [51]. Briefly, 3 mL of SW1353 cells (5 × 10^4^ cells/mL) was seeded in a 6-well plate for 24 h, followed by treatment with H_2_O_2_ and MIA at their IC_25_ and IC_50_ values, or IL-1β at 50 and 100 ng/mL for 24 h. Before collecting the cells, the microscopic images of cells were captured at 200× magnification using an Olympus CKX31 inverted microscope with an X-CAM alpha camera and DigiAcquis 2.0 acquisition software (Matrix Optics, Petaling Jaya, Malaysia) for morphological comparisons. All the treated cells, including the floated cells, were then collected, trypsinised and washed twice with ice-cold PBS at 200 g for 5 min. The cells were resuspended in 350 µL of Annexin V staining buffer (BD Bioscience, San Jose, CA, USA) and stained with 5 µL of Annexin V conjugated with fluorescein isothiocyanate (Annexin V–FITC) (BD Bioscience, San Jose, CA, USA) on ice in dark conditions for 15 min. A 10 µL solution of 50 µg/mL propidium iodide (PI) was then added to the cells and incubated for another 5 min. After the staining, another 150 µL of Annexin V staining buffer was added to the cells, and the entire content was transferred to a flow tube for analysis. At least 10,000 stained cells were recorded and analysed using a BD FACSVerse flow cytometer (BD Bioscience, San Jose, CA, USA). The cell population with negative staining for both markers represents the viable cell population, positive staining for Annexin V–FITC represents the early apoptotic population, positive staining for PI represents the necrotic population, whereas positive staining for both markers represents a late apoptotic population. Both early and late apoptotic populations are collectively reported as the apoptotic population in this study.

### 2.5. Cell Cycle Assay

The cell cycle assay was conducted in accord with a previous study [51]. Briefly, 3 mL of SW1353 cells (5 × 10^4^ cells/mL) was seeded in a 6-well plate for 24 h, followed by treatment with the IC_50_ of MIA for 24 h. A serum-free condition was used as a cell cycle arrest control. All the treated cells, including the floating cells, were then collected, trypsinised and washed twice with ice-cold PBS at 200 g for 5 min. The cells were then counted, and 1 × 10^5^ cells were fixed with 1 mL of 70% ethanol drop by drop while being vortexed. After that, they were kept overnight. The fixed cells were washed again with ice-cold PBS at 400 g for 10 min. Subsequently, the cells were stained with 500 µL of PI/RNase staining solution (BD Bioscience, San Jose, CA, USA) for 15 min at room temperature. At least 12,000 stained cells were recorded, and the cell cycle distribution was further analysed using ModFit LT software (Verity Software House, Topsham, ME, USA) via Gaussian curve modelling estimation [52,53].

### 2.6. Lysate Preparation and Protein Determination

Briefly, 15.5 mL of SW1353 cells (5 × 10^4^ cells/mL) was seeded in a 100 mm culture dish for 24 h, followed by treatment with the IC_25_ and IC_50_ values of H_2_O_2_ and MIA, or IL-1β at 50 and 100 ng/mL for another 24 h. All the treated cells, including the floating cells, were then collected, trypsinised and washed twice with ice-cold PBS at 200 g for 5 min. The cells were then lysed with 200 µL of Qproteome mammalian lysis buffer (Qiagen, Hilden, Germany) with protease inhibitors for 15 min on ice with gentle vortexing from time to time. The contents were then centrifuged at 4 °C and 10,000× *g* for 15 min. The supernatants were collected as lysates and kept at −80 °C until analysis. The protein concentrations of lysates were determined using Bio-Rad Protein Assay concentrated dye reagent (Bio-Rad Laboratories, Hercules, CA, USA) in microtiter format according to the manufacturer’s instructions. Bovine serum albumin (0–2 mg/mL) was used as the protein standard. The absorbance of each well was measured at 595 nm using a Multiskan GO microplate reader (Thermo Fisher Scientific, Vantaa, Finland). All the protein concentrations of lysates (500 µL) were then diluted to 1 mg/mL for the subsequent quantitative detection of isoprostane and TNF-α levels.

### 2.7. Enzyme-Linked Immunosorbent Assay

Human 8-isoprostane F2-α (catalogue No. E4805Hu) and TNF-α (catalogue No. E-EL-H0109) were determined using the sandwich-ELISA kits from Bioassay Technology Laboratory (Shanghai, China) and Elabscience (Houston, TX, USA). The procedures were conducted according to the manufacturer’s instructions [54,55]. For human 8-isoprostane F2-α determination, 40 μL of lysates (1 mg/mL) and 10 μL of human 8-isoprostane F2-α antibody were added in duplicate into a 96-well plate pre-coated with human 8-isoprostane F2-α antibody. A 50 µL aliquot of standard solutions (0–800 ng/L) was added separately into other wells. Next, 50 μL of streptavidin–horseradish peroxidase (HRP) solution was added into all wells (except the blank) and the plate was incubated at 37 °C for 1 h. After that, the plate was washed 5 times with a wash buffer and blot-dried with paper towels, followed by the addition of 100 μL of substrate Solution A and substrate Solution B in a 1:1 ratio. The plate was sealed and further incubated at 37 °C in the dark for another 10 min. Lastly, 50 μL of the stop solution was added into each well, and the absorbance of each well was measured at 450 nm by using a Multiskan GO microplate reader (Thermo Fisher Scientific, Vantaa, Finland). The human 8-isoprostane F2-α levels of the lysates were calculated based on the standard curve.

For human TNF-α quantification, 100 μL of lysates (1 mg/mL) was added in duplicate into a 96-well plate pre-coated with human TNF-α antibody. Similar amounts of standard solutions (0–500 pg/mL) were added separately into other wells. The plate was incubated at 37 °C for 90 min. All the content of wells was then removed and 100 μL of a biotinylated detection antibody working solution was added directly. The plate was further incubated at 37 °C for 1 h. After that, the plate was washed 3 times with a wash buffer and blot-dried with paper towels, then 100 μL of an HRP conjugate working solution was added. The plate was sealed and further incubated at 37 °C for another 30 min. After that, the plate was washed 5 times with a wash buffer and blot-dried with paper towels, then 90 μL of the substrate reagent was added. The plate was incubated at 37 °C for another 15 min. Lastly, 50 μL of the stop solution was added into each well and the absorbance of each well was measured at 450 nm by using a Multiskan GO microplate reader (Thermo Fisher Scientific, Vantaa, Finland). The human TNF-α levels of lysates were determined based on the standard curve.

### 2.8. Statistical Analysis

The data were collected from three independent experiments with the respective technical replicates. Statistical analysis was conducted using SPSS software for Windows, version 25 (IBM, Armonk, NY, USA). Normality tests confirmed the normal distribution of data. Mean differences among the study groups were analysed with one-way analysis of variance with Tukey’s or Dunnett’s test as post hoc analysis. A *p*-value of less than 0.05 was considered statistically significant.

## 3. Results

### 3.1. Cytotoxicity of MIA, H_2_O_2_ and IL-1β on SW1353 Chondrocytes

Figure 1 shows the cytotoxicity of IL-1β, MIA and H_2_O_2_ on SW1353 chondrocytes upon 24 h of treatment. MIA (25 and 50 μM) and H_2_O_2_ (50, 100 and 200 μM) were cytotoxic to SW1353 chondrocytes, with a concentration-dependent decline in cell viability (all *p* < 0.001). The IC_25_ and IC_50_ values for H_2_O_2_ treatment were 55 and 82.5 μM. The values were 22.5 and 31.5 μM for MIA treatment. IL-1β did not cause a significant decline in the viability of cells up to 100 ng/mL upon 24 h treatment (*p* > 0.05). Therefore, the highest and second-highest concentrations of IL-1β (50 and 100 ng/mL) were used in the subsequent experiment.

### 3.2. Morphological Changes of SW1353 Chondrocytes upon MIA, H_2_O_2_ and IL-1β Treatment

Figure 2 shows the morphology of SW1353 chondrocytes after 24 h of MIA, H_2_O_2_ and IL-1β treatment. SW1353 cells in the VC group demonstrated a typical spindle-shaped fibroblast-like morphology. Active and proliferating cells with a shiny appearance were also observed. Consistent with the cytotoxicity and mode of cell death findings, IL-1β did not cause any morphological changes in SW1353 chondrocytes compared with VC. Upon MIA treatment, there was no obvious change in SW1353 cell morphology, but the number of attached cells was reduced in a concentration-dependent manner. On the other hand, SW1353 chondrocytes treated with H_2_O_2_ demonstrated typical apoptotic features, including blebbing formation, cell shrinkage and apoptotic body formation. Besides, H_2_O_2_-treated SW1353 cells exhibited elongated or epithelial-shaped cells with reduced cell density.

### 3.3. Mode of SW1353 Cell Death under MIA, H_2_O_2_ and IL-1β Treatment

Plasma membrane phosphatidylserine externalization is one of the earliest signs of apoptosis [50,56]. The exposed phosphatidylserine on the cell surface can be bound with Annexin V in the presence of calcium ions [57]. In this study, an Annexin V–FITC/PI dual-labelling assay was used to distinguish the mode of SW1353 cell death, either via apoptosis and/or necrosis. Figure 3 summarises the viable, apoptotic and necrotic cell populations with or without the respective treatments. H_2_O_2_ (both IC_25_ and IC_50_) significantly induced SW1353 cell death with a primarily increase in the apoptotic population (*p* < 0.001) and a slight increase in the necrotic population (*p* < 0.001). Parallel with the cytotoxicity data, IL-1β did not induce significant SW1353 cell death compared with the VC (*p* > 0.05). Surprisingly, MIA (IC_25_ and IC_50_) did not induce significant SW1353 cell death after 24 h of treatment (*p* > 0.05).

### 3.4. Cell Cycle Analysis

Figure 4 demonstrates the cell cycle analysis of SW1353 chondrocytes with VC, MIA (IC_50_) and serum-free media for 24 h. Serum deprivation is known to induce G1 arrest [58]. In this study, SW1353 cells with serum-free media showed a significant accumulation of the G1-phase cell population (*p* < 0.001), while the MIA treatment significantly increased the S-phase cell population (*p* < 0.05) with the lower G2-phase population, indicating a S-phase arrest.

### 3.5. 8-Isoprostane F2-α and TNF-α Levels

8-isoprostane F2-α is one of the oxidative stress and lipid peroxidation markers, which is stably formed via non-enzymatic peroxidation of fatty acids, primarily arachidonic acid [59]. Figure 5 shows the 8-isoprostane F2-α and TNF-α levels of SW1353 chondrocytes after 24 h MIA, H_2_O_2_ and IL-1β treatment. The 24 h treatment with MIA and H_2_O_2_ significantly reduced the TNF-α levels (all *p* < 0.05) compared with the VC, while the IL-1β treatment significantly increased the TNF-α levels (all *p* < 0.001) compared with the VC in a concentration-dependent manner. On the other hand, SW1353 chondrocytes treated with the IC_25_ and IC_50_ of H_2_O_2_ demonstrated significantly higher 8-isoprostane F2-α levels (all *p* < 0.05). A similar increasing trend was observed in MIA-treated SW1353 chondrocytes, but only the IC_50_ of MIA treatment showed statistical significance compared with the VC (all *p* < 0.05). Neither 50 nor 100 ng/mL IL-1β significantly altered the isoprostane levels in SW1353 chondrocytes (*p* > 0.05), despite a significant induction of TNF-α levels.

## 4. Discussion

In this study, 24 h of IL-1β was non-cytotoxic to SW1353 chondrocytes, without significant cell death up to 100 ng/mL. The cell morphology of IL-1β-treated SW1353 cells appeared similar to the control, consistent with the cytotoxicity and cell death findings. Besides, IL-1β-treated SW1353 cells were detected at a higher intracellular TNF-α level but insignificant changes to the 8-isoprostane F2-α (oxidative stress marker) level. On the other hand, 24 h of H_2_O_2_ induced concentration-dependent cytotoxicity and cell death, primarily via apoptosis in SW1353 chondrocytes, with the typical apoptotic features including blebbing formation, cell shrinkage and cell debris formation. Besides, there was a slight but significant increase (3–5%) in the necrotic cell population upon H_2_O_2_ treatment which might be due to the direct H_2_O_2_ damage to the cell membrane [35]. H_2_O_2_ also significantly increased the 8-isoprostane F2-α levels in SW1353 cells, which is coherent with its known oxidant activity [15,37,38]. However, H_2_O_2_ reduced the intracellular TNF-α levels in SW1353 chondrocytes. On the other hand, 24 h of MIA decreased the viability of SW1353 cells in a concentration-dependent manner but without significant cell death. MIA-treated SW1353 chondrocytes were morphologically similar to the VC but with fewer attached cells. Subsequently, we demonstrated that MIA treatment caused S-phase cell cycle arrest on SW1353 cells, with a higher level of 8-isoprostane F2-α. DNA damage and GSH depletion have been reported to cause S-phase arrest for cell repair process [60]. Therefore, it is rational to speculate that MIA-mediated oxidative stress may induce oxidative DNA damage to SW1353 cells, which subsequently triggers cell cycle arrest but not cell death.

The proinflammatory cytokine IL-1β was used to induce inflammation in SW1353 chondrocytes at the concentrations of 5–20 ng/mL, but its cytotoxic effect was not determined [61,62,63,64,65,66,67,68,69,70]. Non-cytotoxic effects of IL-1β (5–10 ng/mL) on SW1353 chondrocytes were evident up to 72 h of treatment [26,71,72,73,74]. Higher (40 ng/mL) or long-term (up to 5 days) treatment of IL-1β also did not affect the viability of SW1353 chondrocytes [73], CHON-002 cells [20] and primary human chondrocytes [75]. Our study demonstrated the non-cytotoxic properties of IL-1β, which is in agreement with previous studies. On the contrary, several studies demonstrated the cytotoxicity of IL-1β on SW1353 chondrocytes, whereby 24 h of IL-1β (10 ng/mL) could reduce the viability to 50–60% [28,29]. Coincidentally, these studies demonstrating the greater cytotoxicity of IL-1β cultured the SW1353 chondrocytes in a serum-free condition before IL-1β treatment [28,29]. Serum deprivation is commonly used in cytokine studies to reduce the background stimulation from the serum components [76]. However, serum deprivation could further enhance the susceptibility towards lipopolysaccharides [77] and cytokines, including IL-1β [78], and also induce cell cycle arrest [58], autophagy and apoptosis [79]. In this study, we did not perform the serum deprivation procedure prior to IL-1β treatment, similar to most of the studies on IL-1β-treated SW1353 cells [26,71,72,73,74].

There was a significant increase in intracellular TNF-α levels in IL-1β-treated cells. Mechanistically, IL-1β activates the activator protein-1 and nuclear factor kappa-light-chain-enhancer of activated B cell signalling pathway, subsequently upregulating the expression of downstream cytokines, including IL-1β and TNF-α [80]. A 24 h of 10 ng/mL IL-1β treatment was reported to increase TNF-α levels by four-fold in serum-starved SW1353 chondrocytes [28]. Only a 1.7- to 3.2-fold increase in TNF-α levels upon 50 and 100 ng/mL of IL-1β treatment was observed in our study. As suggested earlier, serum deprivation may enhance the IL-1β signalling in SW1353 cells and greatly upregulated the TNF-α level, despite being at lower concentration. In terms of morphology, the IL-1β- or lipopolysaccharide-stimulated primary human articular chondrocytes displayed no noticeable changes [81] or changed to a rounded shape with smaller cell size [75]. In both cases, no cell death features were observed. Similarly, we also demonstrated that IL-1β did not induce obvious morphological changes in SW1353 cells.

Oxidative stress occurs when cell antioxidant defences are overwhelmed by ROS accumulation [82]. Previous studies reported that IL-1β (40 ng/mL) could induce ROS production in chondrocytes [73]. However, a time-course analysis on the 10 ng/mL IL-1β-treated SW1353 chondrocytes revealed that ROS levels were significantly increased only during the first hour of treatment [83]. Subsequently, the ROS levels reverted to baseline when the intracellular superoxide dismutase level was adaptively upregulated due to redox signalling [83]. Moreover, a non-cytotoxic ROS level could act as a secondary messenger to trigger redox and cytokine signalling [84,85]. Only overwhelmingly high ROS levels could damage the cell antioxidant defence, resulting in oxidative damage and cell death [86]. In this study, IL-1β might upregulate an early but non-lethal ROS level as a secondary messenger to stimulate the cytokine signalling, without overwhelming the cell antioxidant defence mechanism. Therefore, cell cycle arrest or cell death events were not triggered. Analysis of the antioxidant levels and/or direct ROS measurement in a time-course manner will be helpful for better visualisation of the oxidative status upon IL-1β treatment.

Our study showed that H_2_O_2_ and MIA were cytotoxic and cytostatic to SW1353 chondrocytes, respectively. These changes were accompanied by an increase in 8-isoprostane F2-α levels but a lower level of intracellular TNF-α. Previous studies showed that the IC_50_ values of H_2_O_2_ in SW1353 chondrocytes ranged from 0.5 to 1 mM upon 1 h to 24 h of treatment [15,37,38,39], while IC_50_ values around 5–8.26 µM for MIA were reported for SW1353 chondrocytes [16,43] and primary chondrocytes [17,46,87]. Our data demonstrated that IC_50_ values of H_2_O_2_ and MIA were 82.5 µM (6- to 12-fold lower) and 31.5 µM (4- to 6-fold higher), respectively. Multiple factors, such as batch-to-batch variation of cells and reagents, and culturing and treatment conditions, may have contributed to this discrepancy. The differences in media composition [88] may affect the sensitivity of cells. The DMEM (ThermoFisher Scientific, catalogue No. 11965092) used in this study contains high glucose and 4 mM l-glutamine without sodium pyruvate. l-glutamine in the media could contribute to the higher IC_50_ value of MIA because it could provide an alternative energy source to the cells when glycolysis is inhibited [89]. Additionally, the greater sensitivity of SW1353 cells towards H_2_O_2_ in our study might be due to the absence of pyruvate in the culture media, where 0.5–1 mM pyruvate (the typical concentration in media) was shown to protect the cells from exogenous H_2_O_2_ [90,91]. Besides, the IC_50_ values of H_2_O_2_ from the media without pyruvate was about five-fold lower than in those containing pyruvate [91], which is similar to our findings. Nevertheless, detailed information on the media composition used in previous studies [15,16,17,37,38,39,43,46,87] is not available. A further comparative study is required to confirm our speculation.

Both H_2_O_2_ and MIA treatment significantly reduced the TNF-α levels in SW1353 chondrocytes despite higher 8-isoprostane F2-α levels. H_2_O_2_ [15,37,38] and MIA [16,17,43,46,87,92] were consistently reported to induce oxidative stress in SW1353 chondrocytes and/or primary rat chondrocytes. However, few studies demonstrated the upregulation of intracellular TNF-α levels in H_2_O_2_- or MIA-treated SW1353 chondrocytes. The available studies showed that 24 h of H_2_O_2_ (0.5–10 mM) increased the release of TNF-α in SW1353 chondrocytes [37] and CHON-001 chondrocytes [19]. A lower concentration of H_2_O_2_ treatment (40 ng/mL or 1.17 µM) also increased the intracellular TNF-α levels in primary rat articular chondrocytes [93]. MIA was reported to upregulate the TNF-α mRNA level [43,92] and secretion of TNF-α [92] in SW1353 chondrocytes. The underlying reason for this discrepancy is unclear. We hypothesise that the induction of cytostasis and apoptosis by MIA and H_2_O_2_ may interfere with the expression of TNF-α. Several proteinases, including caspases and calpains, are activated during apoptosis and they actively degrade the intracellular proteins and organelles [94]. Further studies are required to confirm the role of the cytostatic and cytotoxic effects of MIA and H_2_O_2_ treatment in downregulating TNF-α levels.

There are several limitations to this study. The use of single oxidative and inflammatory markers renders the interpretation of data non-comprehensive. The status of other cytokines or mediators like IL-6 and MMPs may be useful in validating the effects of OA inducers. Besides, direct measurement of ROS species is more informative than an indirect indication from the oxidative markers, although the ROS are extremely reactive and transiently present, which may be difficult to visualize [95]. Time-course analysis of the ROS levels and inflammatory markers may help to identify the role of redox and cytokine signalling in mediating the effects of these inducers in chondrocytes. Moreover, the single time-point assessment used in this study may have missed out some critical changes in cell growth.

## 5. Conclusions

In summary, our study demonstrated that H_2_O_2_ induced apoptosis while MIA induced a cytostatic effect on SW1353 cells, without a significant inflammation process. IL-1β is a suitable inducer of inflammation but not SW1353 cell death. Caution should be taken regarding the treatment conditions, especially the media composition and serum-free conditions in these in vitro OA models.

## Figures and Tables

**Figure 1 life-11-00272-f001:**
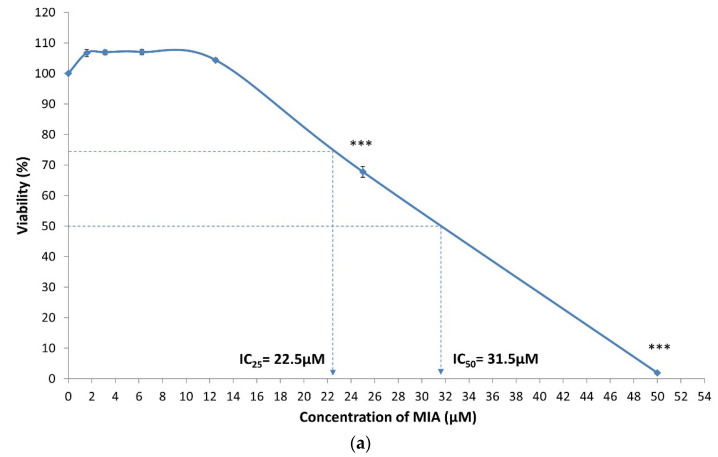
Monosodium iodoacetate (MIA), H_2_O_2_ and interleukin-1β (IL-1β) exerted differential effects on cytotoxicity of SW1353 chondrocytes. (**a**) A 24 h treatment of MIA induced cytotoxicity on SW1353 chondrocytes in a concentration-dependent manner with 25% and 50% inhibitory concentration (IC_25_ and IC_50_) values of 22.5 μM and 31.5 μM, respectively. (**b**) A 24 h treatment of H_2_O_2_ also induced concentration-dependent cytotoxicity on SW1353 chondrocytes. H_2_O_2_ was less potent than MIA, with greater IC_25_ and IC_50_ values of 55 μM and 82.5 μM. (**c**) IL-1β did not cause any cytotoxic effect on SW1353 chondrocytes after 24 h of treatment. Viability was calculated as a percentage using the optical density (OD) of the vehicle control (VC) as the denominator. Three independent experiments were conducted with four technical replicates. *** *p* < 0.001 against the VC.

**Figure 2 life-11-00272-f002:**
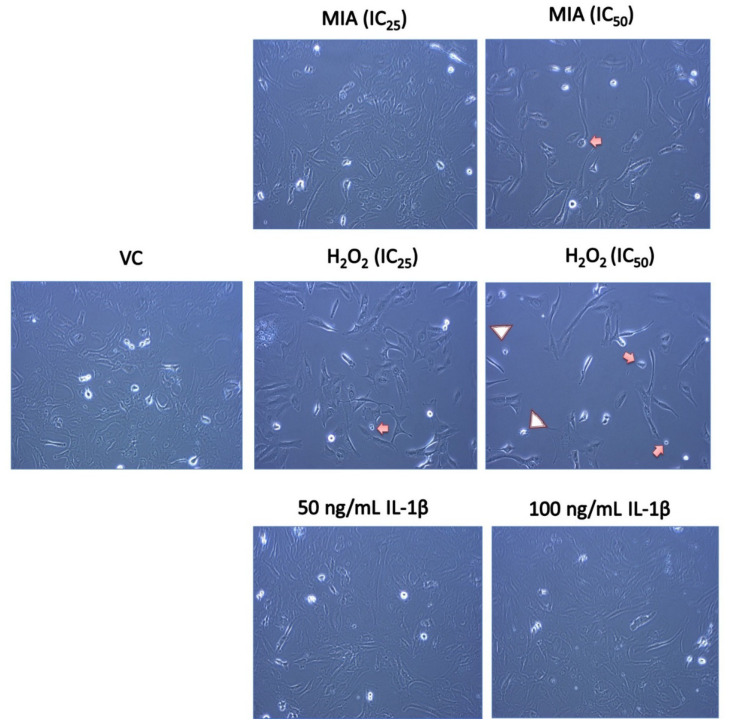
Morphological changes of SW1353 chondrocytes upon MIA, H_2_O_2_ and IL-1β treatment. Chondrocytes in the VC group had a typical spindle-shaped cell morphology. Several round and shiny proliferating cells were observed. MIA treatment did not cause any obvious change in cell morphology but concentration-dependently reduced the number of attached cells. Chondrocytes treated with H_2_O_2_ showed the typical apoptotic blebbing formation (arrowhead), cell shrinkage (arrow) and cellular debris, with a substantial reduction in cell number. IL-1β did not cause any morphological changes in chondrocytes compared with the VC. Representative images from three independent experiments are shown with 200× magnification.

**Figure 3 life-11-00272-f003:**
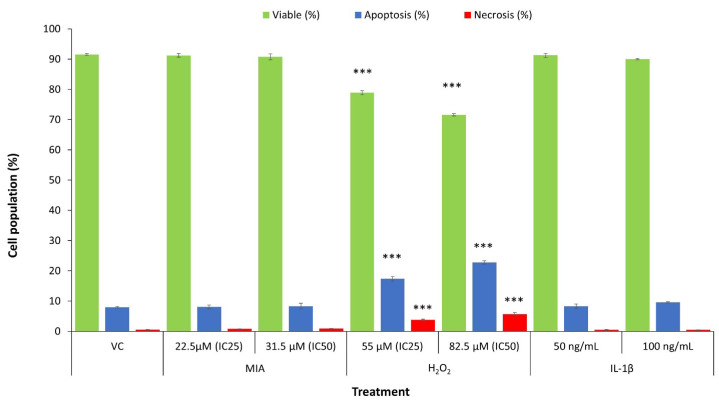
Mode of SW1353 cell death under treatment with MIA, H_2_O_2_ and IL-1β. The IC_25_ and IC_50_ of MIA did not significantly induce apoptosis or necrosis in SW1353 chondrocytes. Therefore, MIA was cytostatic at these concentrations, whereby it arrested the growth of SW1353 chondrocytes but did not induce cell death. On the other hand, H_2_O_2_ (IC_25_ and IC_50_) significantly induced apoptosis in SW1353 chondrocytes. Additionally, H_2_O_2_ marginally but significantly increased necrosis in chondrocytes. IL-1β (50 and 100 ng/mL) did not significantly induce apoptosis or necrosis in SW1353 chondrocytes, parallel with the cytotoxicity result. The data were collected from three independent experiments. *** *p* < 0.001 against the VC.

**Figure 4 life-11-00272-f004:**
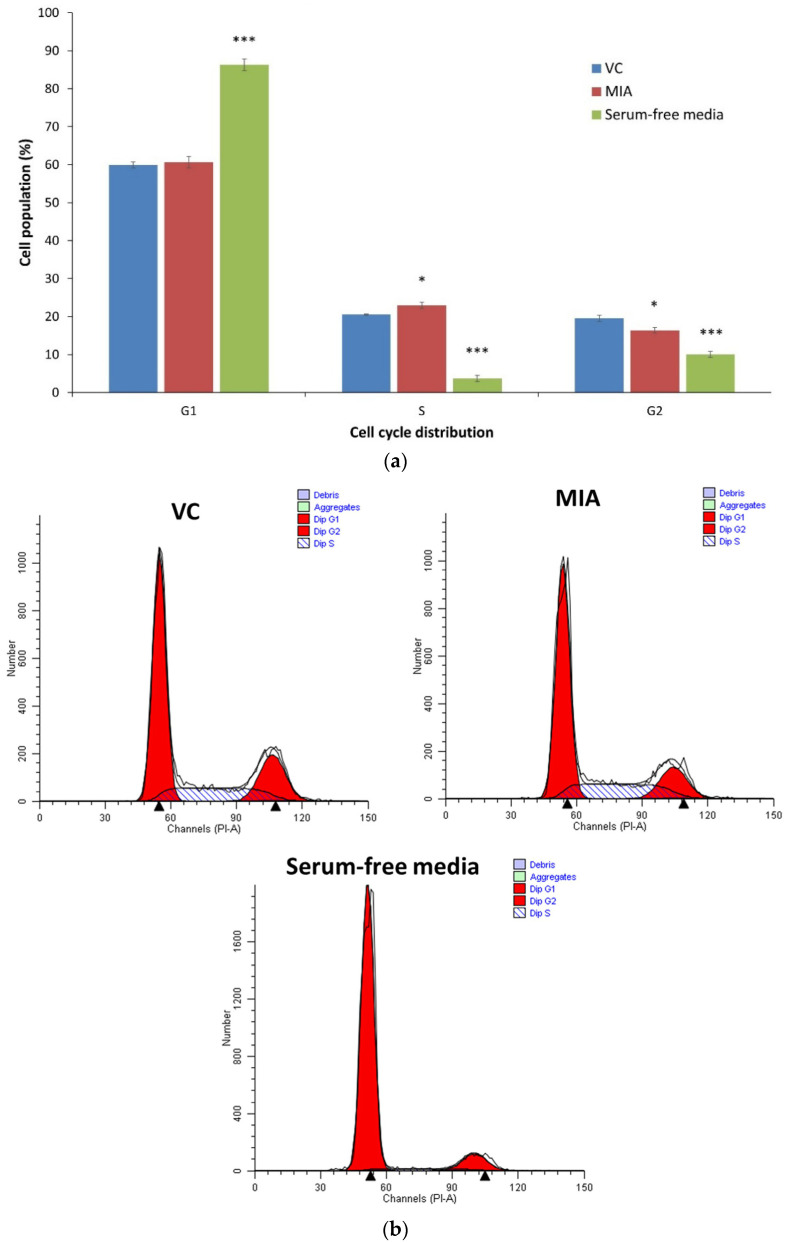
Quantitative analysis of cell cycle distribution after MIA treatment. (**a**) MIA (IC_50_) induced S-phase arrest in SW1353 chondrocytes. Serum-free conditions arrested the SW1353 cells at the G1 phase. The data were collected from three independent experiments. (**b**) Representative cell cycle distribution analysis from the ModFit LT software. * *p* < 0.05 against the VC; *** *p* <0.001 against the VC.

**Figure 5 life-11-00272-f005:**
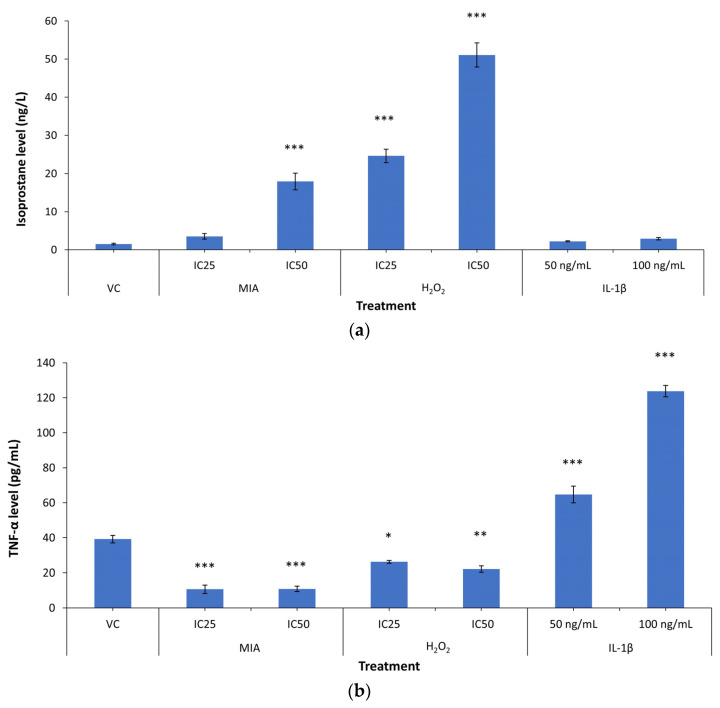
The isoprostane and tumour necrosis factor-α (TNF-α) levels under MIA, H_2_O_2_ and IL-1β treatment. (**a**) H_2_O_2_ and MIA (IC_50_ only) induced oxidative stress, with significantly higher isoprostane levels compared with the VC. IL-1β did not cause any upregulation of isoprostane levels, indicating the absence of oxidative stress. (**b**) IL-1β induced inflammation with higher TNF-α levels. H_2_O_2_ and MIA did not induce inflammation, with low TNF-α levels. The data were collected from three independent experiments with two technical replicates. * *p* < 0.05 against the VC; ** *p* < 0.01 against the VC; *** *p* < 0.001 against the VC.

## Data Availability

The data presented in this study are available on request from the corresponding author. The data are not publicly available due to institutional policy.

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
