# Peer review of "Establishing SW1353 Chondrocytes as a Cellular Model of Chondrolysis"

_life, 2021, doi:10.3390/life11040272_

Round 1

Reviewer 1 Report

Manuscript 1125979:

Establishing SW1353 chondrocytes as a cellular model of chondrolysis

(Authors: Kok-Lun Pang, Yoke Yue Chow, Lek Mun Leong, Jia Xian Law, Norzana Abd Ghafar, Ima Nirwana Soelaiman, Kok-Yong Chin)

In this manuscript authors studied effects of hydrogen peroxide (H2O2), glycolysis inhibito monosodium iodoacetate (MIA), and interleukin-1β (IL-1β) on cell death of human chondrosarcoma cell line SW1353. Oxidative stress, measured by isoprostane level, and decrease in cell viability were detected after H2O2 and MIA treatment. H2O2 induced apoptosis while MIA induced S-phase arrest. IL-1β induced intracellular TNF-a, while H2O2 and MIA did not. IL-1b did not change the isoprostane level. Authors conclude that IL-1b is suitable for inflammation studies, H202 for cell death studies while MIA for the growth arrest studies.

Minor comments:

Page 1, line 4 and 5: Name of the author should not be cut “Ima Nirwana Soelaim-an” should be replaced withIma Nirwana Soelaiman”

Page 2, line 84: “SW1353 chondrocyte” should be replaced with “Chondrosarcoma cell line SW1353”.  

In Figure legend for Figure 3 (Page 9, line 317): “did not kill them” should be replaced with “did not induce cell death.”

Author Response

Dear reviewer, 

Thank you for reviewing our manuscript. Your constructive comments are much appreciated and responded in the attached response sheet. 

Thank you. 

Reviewer 2 Report

The manuscript by Pang et al describes in great detail the effect of osteoarthritis relevant stimuli IL1b, MIA and H202 on a commonly used chondrocytic cell-line SW1353. The study is well executed and the manuscript well written. A major point of criticism is the overwhelming number of references for a straightforward experimental study (154, where one would realistically expect at least half this number or less then fifty). Another concern is that the novelty is not so high. Also, one can question the relevance of IL1b and TNFa in osteoarthritis given recent failure of clinical trials with neutralizing antibodies. However, these signaling molecules are often used as a model in literature and the authors nicely compare their results in the discussion section to similar studies and find some discrepancies that have scientific merit. In conclusion, for the osteoarthritis community publication of this data is relevant. However, in the opinion of the reviewer the number of references should be drastically reduced.

Author Response

(The authors gave the same response as above.)

Reviewer 3 Report

The current manuscript lacks scientific novelty or originality, and its content is of no particular significance to the field.

The role of OA inducers has been extensively studied, including in the current cell line. furthermore, the current manuscript does not add any new scientific knowledge to the field.

Author Response

(The authors gave the same response as above.)

Reviewer 4 Report

This manuscript is well-written and informative. However, there are still some suggestions and concerns, 
1. INTRODUCTION. Background is too much. Suggest to delete the first paragraph. Only focusing on the center of topic is enough. Please concise this part. 
2. Please identify the rationale of this study in the INTRODUCTION.
3. Overall,  references are reach above 150, that is too much. Kindly choose the main references and narrow down to a number of 60. 
4.  In this study, passage number seems to higher (28-36). Did it result in the different outcome? Please discuss
5. Please provide the sample number (n=?) and repeated ? times for each assay. 
6. Figure 1. Why viability > 100%? 
7. Figure 2. Provide the scale bar. 
8. Figure. Please clarify  how to determine and calculate the % of apoptosis and necrosis on the basis of what? In addition, suggest to provide the IHC staining image of biomarkers (ie., caspase-3 for apoptosis,.....for necrosis

, respectively), if possible.  
9. Figure 4. the digits and explanation in the each figure is not easy to read
10.  why ONLY chose 24h for assessment of time point? It would be missed some critical changes of cell growth. 

Author Response

(The authors gave the same response as above.)

Round 2

Reviewer 3 Report

................................................................................................................................................................................................................................................................................................................................................................................................................................................................................................